# Do people with disabilities experience disparities in cancer care? A systematic review

**Irene Tosetti**[1]*, **Hannah Kuper**[2]

**1** M.Sc. Public Health, London School of Hygiene & Tropical Medicine, London, United Kingdom,
**2** International Centre for Evidence in Disability, London School of Hygiene & Tropical Medicine, London, United Kingdom

* itosetti@gmail.com

**Data Availability Statement:** All relevant data are within the paper and its Supporting information files.

**Funding:** This study was financially supported by the National Institute for Health Research (NIHR) in the form of a grant.

## Abstract

### Background

Over 1.3 billion people, or 16% of the world's population, live with some form of disability. Recent studies have reported that people with disabilities (PwD) might not be receiving state-of-the-art treatment for cancer as their non-disabled peers; our objective was to systematically review this topic.

### Methods

A systematic review was undertaken to compare cancer outcomes and quality of cancer care between adults with and without disabilities (NIHR Prospero register ID number: CRD42022281506). A search of the literature was performed in July 2022 across five databases: EMBASE, Medline, Cochrane Library, Web of Science and CINAHL databases. Peer-reviewed quantitative research articles, published in English from 2000 to 2022, with interventional or observational study designs, comparing cancer outcomes between a sample of adult patients with disabilities and a sample without disabilities were included. Studies focused on cancer screening and not treatment were excluded, as well as editorials, commentaries, opinion papers, reviews, case reports, case series under 10 patients and conference abstracts. Studies were evaluated by one reviewer for risk of bias based on a set of criteria according to the SIGN 50 guidelines. A narrative synthesis was conducted according to the Cochrane SWiM guidelines, with tables summarizing study characteristics and outcomes. This research received no external funding.

### Results

Thirty-one studies were included in the systematic review. Compared to people without disabilities, PwD had worse cancer outcomes, in terms of poorer survival and higher overall and cancer-specific mortality. There was also evidence that PwD received poorer quality cancer care, including lower access to state-of-the-art care or curative-intent therapies, treatment delays, undertreatment or excessively invasive treatment, worse access to in-hospital services, less specialist healthcare utilization, less access to pain medications and inadequate end-of-life quality of care.

**Competing interests:** The authors have declared that no competing interests exist.

## Discussion

Limitations of this work include the exclusion of qualitative research, no assessment of publication bias, selection performed by only one reviewer, results from high-income countries only, no meta-analysis and a high risk of bias in 15% of included studies. In spite of these limitations, our results show that PwD often experience severe disparities in cancer care with less guideline-consistent care and higher mortality than people without disabilities. These findings raise urgent questions about how to ensure equitable care for PwD; in order to prevent avoidable morbidity and mortality, cancer care programs need to be evaluated and urgently improved, with specific training of clinical staff, more disability inclusive research, better communication and shared decision-making with patients and elimination of physical, social and cultural barriers.

## Introduction

Cancer is a leading cause of death worldwide, resulting in nearly ten million deaths in 2020 according to WHO data [1]. In spite of this enormous burden of disease, late-stage presentation and lack of diagnosis and treatment remain common, leading to much higher mortality rates [2]. Each cancer type requires a different treatment regimen, so a correct diagnosis is essential to receiving the best treatment and reducing mortality [1, 3]. Good quality of care can also improve quality of life (e.g. through pain management), even when cure is no longer possible. Access to appropriate treatment is therefore of crucial importance, but inequalities in access have been observed for several groups, including PwD [4–10]. Over 1.3 billion people, or 16% of the world's population, live with some form of disability, according to the 2022 World Report on Disability [11]. This figure is expected to grow further in the coming decades, as the population ages and chronic health conditions increase globally. On average, PwD are more likely to experience poor health, because of their underlying health condition/impairment and their socio-economically excluded position in society [12, 13]. They also face a range of barriers to accessing care, including long waiting times, high costs, ableist discrimination by health professionals, inaccessible buildings, inconvenient locations, and lack of communication among different parts of the healthcare team [12–14]. As a consequence, unmet healthcare needs are greater for PwD, contributing towards poorer health and higher mortality [11]. This general pattern of disability-related healthcare exclusion is reflected in known disparities in the use of cancer prevention services, as PwD have lower cancer screening rates than those without disabilities [13–18]. This gap may also exist with respect to cancer care, as several studies have recently reported that patients with disabilities might not be receiving state-of-the-art treatment standards for their cancers [19–23]. Furthermore, several studies suggest that cancer may be diagnosed at a later stage in patients with disabilities, and that they experience treatment disparities resulting in higher cancer-specific mortality rates [24, 25]. A recent meta-analysis from the USA showed that women with disabilities have 0.78 (95% CI: 0.72–0.84) lower odds of attending breast cancer screening and have 0.63 (95% CI: 0.45–0.88) lower odds of attending cervical cancer screening, compared to women without disabilities. A recent study from Taiwan reported that the probability of receiving colorectal cancer screening in people in the four categories of disability (intellectual and developmental disability, dementia, multiple disabilities, and moving functional limitation; OR = 0.53, 0.55, 0.62 and 0.81, respectively) was significantly lower than that in the general population [24, 26]. Two recent scoping reviews found that patients with intellectual disabilities may be at risk of experiencing

inequities at various points during cancer clinical pathways, which as a consequence could have an impact on their overall and cancer-specific mortality and quality of life; it is thus of the outmost importance to identify and address these disparities [25, 27]. Consequently, the aim of this study is to conduct a systematic literature review to compare cancer outcomes and quality of cancer care between adults with and without disabilities.

## Materials and methods

A systematic review of the literature was conducted describing differences in cancer-related care between patients with and without disabilities, according to the PRISMA reporting guidelines; the study was recorded on the NIHR Prospero register of systematic reviews with ID number CRD42022281506 [28, 29].

### Search strategy

We used a systematic literature review to achieve our aim and objectives. The review was performed on July 1st 2022, across 5 databases: EMBASE, Medline, Cochrane Library, Web of Science and CINAHL databases. We included search terms on: disability (physical, sensory, psychological, communication and/or cognitive disability; measured clinically or through self-report); and cancer treatment (surgery, radiotherapy, chemotherapy, palliative care for any type of cancer), limited to the past 22 years (2000-June 2022), and to English language because of resource challenges with respect to costs, time, and expertise in non-English languages. The full search strategy can be found in the S1 File.

Eligible studies included quantitative studies (observational or interventional), conducted in adults aged 18+, allowing comparison of cancer outcomes between PwD (of any type) and those without disabilities. The disability definition had to be in agreement with the International Classification of Functioning, Disability and Health (ICF) framework (i.e. including impairment, activity limitations or participation restriction due to an underlying health condition in interaction with personal and environmental barriers) [30]. Studies had to include one or more measures of outcomes along the cancer clinical pathway of diagnosis, treatment, and follow-up or end-of-life care. Eligible outcomes were overall mortality after cancer diagnosis, cancer-related mortality, survival, access to state-of-the-art treatment (defined as intent-to-cure treatment when feasible or guideline-consistent stage-appropriate treatment), type of treatment received (medical vs surgical vs radiation vs. hormonal), invasiveness of treatment, delay of treatment, specialist care utilization, access to pain control prescription and end-of-life hospital use for palliative care. Studies focused on screening for cancer were not eligible, as this question was recently reviewed [15, 24, 27, 30–32]. There were no geographic restrictions.

Types of study excluded were editorials, commentaries, opinion papers, reviews, case reports, case series under 10 patients and conference abstracts. Studies with patients under age 18 in a paediatric setting, studies without a measure of disability, studies that did not include a sample of patients with disabilities and a sample of patients without disabilities and studies without outcome measures for cancer care were also excluded. According to these criteria, studies with ineligible design, comparator, population, outcomes, intervention or setting were excluded.

### Study selection

All studies identified through the searches were exported to a Mendeley bibliographic database for deduplication and to Covidence software for screening. One author (IT) screened studies by title and abstract and full text to determine eligibility. Decisions to include were made according to inclusion criteria.

## Data extraction and analysis

A table was created for data extraction (S1 Table in S1 File) listing authors, year of publishing, country where the study was undertaken, study design, type of cancer, type of disability, type of outcome, population size and overall risk of bias. One author (IT) extracted the data. A summary of study characteristics can be found in Table 1.

We also created a summary of primary and secondary outcomes of each study); where possible, odds or prevalence ratios as a measure of association or *p*-values comparing measures in people with and without disabilities were extracted. Each study was also classified as "better", "worse" or "null", when outcomes respectively showed a better, worse or equal situation in quality of cancer care for PwD in comparison to people without disabilities.

A narrative synthesis was conducted according to the Cochrane SWiM guidelines.

## Determining risk of bias

Studies were evaluated for risk of bias based on a set of criteria according to the SIGN 50 (Scottish Intercollegiate Guidelines Network) checklists as mentioned in S1 File [33].

**Table 1. Summary of characteristics of included studies.**

| | | N | % |
|---|---|---|---|
| REGION (as per WHO classification) | Western Pacific | 14 | 45% |
| | European | 11 | 36% |
| | Americas | 9 | 29% |
| | African | 0 | 0 |
| | South East Asian | 0 | 0 |
| | Eastern Mediterranean | 0 | 0 |
| STUDY DESIGN | Retrospective cohort | 27 | 87% |
| | Prospective cohort | 3 | 10% |
| | Cross-sectional | 1 | 3% |
| DISABILITY TYPE | Visual | 0 | 0% |
| | Hearing | 0 | 0% |
| | Physical | 0 | 0% |
| | Intellectual-cognitive | 9 | 29% |
| | Psychosocial | 13 | 42% |
| | All types | 9 | 29% |
| SAMPLE SIZE OF PEOPLE WITH DISABILITY | Smallest | 46 | n/a |
| | 25th percentile | 523 | n/a |
| | Median | 1016 | n/a |
| | 75th percentile | 4077 | n/a |
| TYPE OF CANCER | Any | 7 | 23% |
| | Breast | 9 | 29% |
| | Stomach and colorectal | 4 | 13% |
| | Lung | 3 | 10% |
| | Prostate | 2 | 6% |
| | Others (Testicular, Multiple Myeloma, Acute Myeloid Leukaemia, Bladder, Oral) | 5 | 16% |
| | All types | 1 | 3% |
| RISK OF BIAS | Low | 12 | 39% |
| | Medium | 14 | 45% |
| | High | 5 | 16% |

Overall ratings were summarised as follows with RobVis tool: [34].

Low risk of bias: all or almost all of the above criteria were fulfilled, and those that were not fulfilled were thought unlikely to alter the conclusions of the study;

- Medium risk of bias: some of the above criteria were fulfilled, and those not fulfilled were thought unlikely to alter the conclusions of the study;

- High risk of bias: few or no criteria were fulfilled, and those that were not fulfilled were thought likely or very likely to alter the conclusions of the study. We did not perform tests to measure publication bias [35].

### Ethical considerations

Approval for the review was given by LSHTM MSc Ethics Board (internal ref. 26741). There were no ethical concerns for this literature review.

## Results

The search was conducted on July 13[th], 2022 resulting in 4140 titles identified (Fig 1).

After removal of 408 duplicates, 3732 titles and abstracts were screened, and 3680 ineligible studies were excluded. Next, 52 full texts were retrieved and 21 were excluded because of ineligible study design, comparator, patient population, outcomes, intervention or setting. Finally, 31 studies were identified as eligible for the systematic review.

### Study characteristics

Table 1 shows a summary of the characteristics of the 31 studies included in the systematic review. All the studies were conducted in high-income countries, with the greatest proportion coming from the USA (29% of the studies, n = 9), followed by South Korea (19%, n = 9), Japan (13%, n = 4), France (10%, n = 3), then by the UK, Netherlands and Sweden with two studies each, and by Belgium, Taiwan and Germany with one study each. Twenty studies (65%) were published after 2018, showing a marked growth in research interest on this topic in the past few years; only 11 eligible studies were published earlier, between 2000 and 2017.

A more detailed table of study characteristics is included in S1 Table of S1 File.

### Study design

Twenty-seven of the 31 studies used a retrospective cohort study design, with data either from a single center (n = 2) or from a national or multi-center health insurance and disability database (n = 25), while three studies used a prospective cohort design, one from a single center and two from multi-center hospital networks. One study only used a cross-sectional design with a survey performed among patients of a network of cancer centers.

### Types of disabilities

Over a third of the eligible studies focused on people with psychosocial disability (42%, n = 13) defined as a previous diagnosis of psychiatric or mental health issues [36–48]. Nine studies (29%, n = 9) focused on intellectual, learning disabilities, cognitive impairment or dementia [49–57]. Nine other studies (29%, n = 9) considered all disability in general or grouped into subcategories (e.g. physical/communication/mental/internal organ/others) [19, 22, 23, 55, 58–62]. Few studies differentiated by severity of impairment [19, 22, 60, 62, 63].

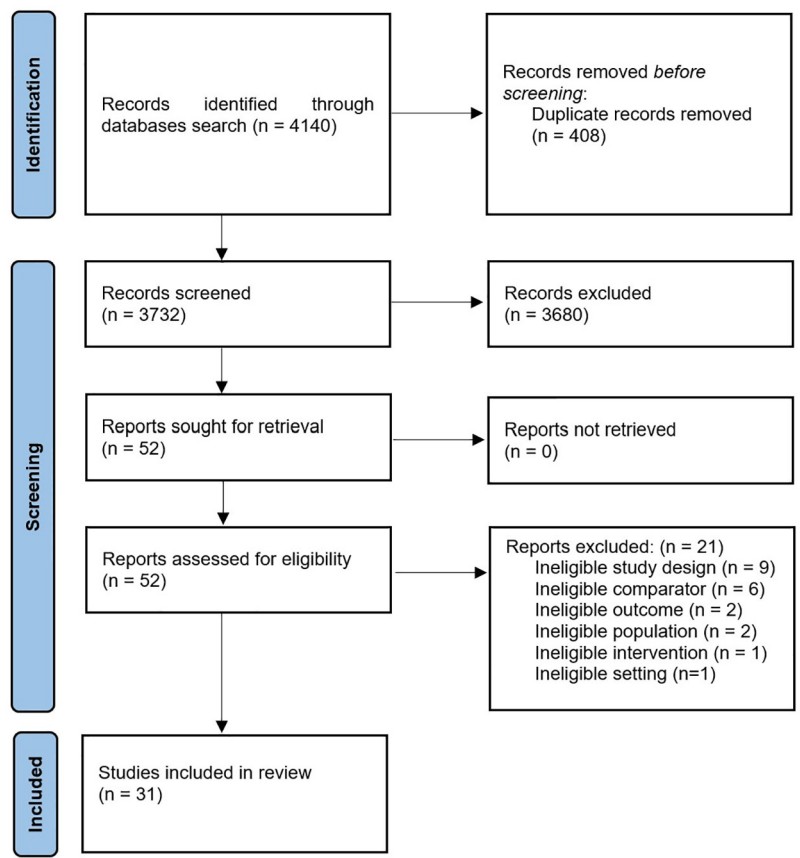

**Fig 1. PRISMA flowchart.**

## Types of cancer

Seven studies (23%) were about any type of malignancy, while almost a third (29%, n = 9) were about breast cancer. There were 4 studies (13%) regarding stomach and colorectal malignancies, 3 (10%) on lung cancer, 2 (6%) on prostate cancer, and 1 study each for testicular, MM, multiple myeloma (MM), acute myeloid leukemia (AML), bladder and oral cancer. Finally, one study included patients of breast, prostate and colorectal cancers (Table 1).

## Types of outcome

The majority of papers (65%, n = 20) included a measure of survival or mortality after cancer diagnosis as primary or secondary outcome. Seventeen studies (55%) included an outcome of access to state-of-the-art cancer treatment, measured as type of treatment received (guidelines consistent according to disease stage) or invasiveness of surgery or treatment delay. Four (13%) studies described access to quality of end-of-life care, defined as access to appropriate pain control and end-of-life hospital use for palliative care. One study included access to pain medications as an outcome.

## Risk of bias

Almost half of the 31 studies (45%, n = 14) had a medium risk of bias, while 12 studies had a low risk of bias (39%, n = 12). Finally, 5 papers were marked as having a high risk of bias. The ratings of risk of bias were summarized (Fig 2) with RobVis tool [36].

## Outcome results

Outcomes are summarized in Table 2.

Nineteen studies that included a measure of survival or mortality all showed, invariably, a direction of effect towards worse outcomes for PwD; this was often worsened by the degree of severity of disability [19, 22, 36, 44, 49, 54, 57, 66]. Only one study found no difference in over-all survival or disease-free survival between patients with and without disabilities [38]. Among psychosocial disabilities, schizophrenia had generally the worst prognosis [36, 39, 42, 48, 60]. In studies that examined survival in cancer patients with all types of disability, there seemed to be far worse outcomes for those with severe disabilities and with intellectual impairment. In one study results showed that patients with schizophrenia had a cancer specific mortality rate 50% higher than patients without disabilities [39]. In another study about bladder cancer, the risk of cancer specific death was 35% higher for patients with severe mental illness compared to people without disabilities [19, 22, 42, 62].

In a large study about gastric cancer patients in South Korea, PwD were more likely not to receive proper staging tests to establish an appropriate treatment plan. Observing subgroups by disability type, the fact of not receiving treatment was more common for people with communication impairment (36.9% in severe and 31.4% in mild communication disability); the authors concluded that disability itself should not be a contraindication for receiving cancer treatment [22]. Another study about patients with leukaemia described how the treatment rate was lowest in those with major internal organ and communication disabilities; while for patients with major internal organ disabilities it is understandable to have a low treatment rate due to vital functions often lacking functional reserve, communication disabilities are not directly related to vital functions and the decision not to treat was hence not based solely on medical factors [63].

There were 16 studies showing lower chance in receiving state-of-the-art cancer treatment for PwD, and only one study with high risk of bias showed no difference, but data about gender and degree of disability was missing [59].

The studies showed that cancer treatment was suboptimal for PwD in many ways, and in particular that they had a lower likelihood of undergoing guideline-consistent surgery when indicated [22, 40, 42, 54, 58, 61, 63, 65]. Several studies showed that when PwD were correctly treated with guideline-consistent surgery, their mortality was similar or only slightly higher than controls [55, 60]. PwD were also more likely to face diagnosis and treatment delays—but not when access to screening was optimal, underlining the importance of good screening access [22, 45–47, 50, 67]. PwD were also less likely to receive curative-intent transplants for blood cancers, and more likely to receive inappropriate radical mastectomy instead of guide-line-consistent minimally invasive procedures for breast cancer [19, 38, 57, 58, 62].

As for end-of-life and palliative care, 4 studies showed a direction of effect towards worse outcomes for PwD [36, 47, 48, 52]. One of these studies, with low risk of bias, showed an asso-ciation between receiving outpatient treatment from a mental health professional and having less end-of-life ED visits, suggesting the importance of access to mental health services to improve end-of-life care [47]. One study showed an association between disability and worse access to prescriptions for pain treatment during cancer care, a situation likely to severely compromise quality of life [53]. Finally, a study reported that patients over age 55 with

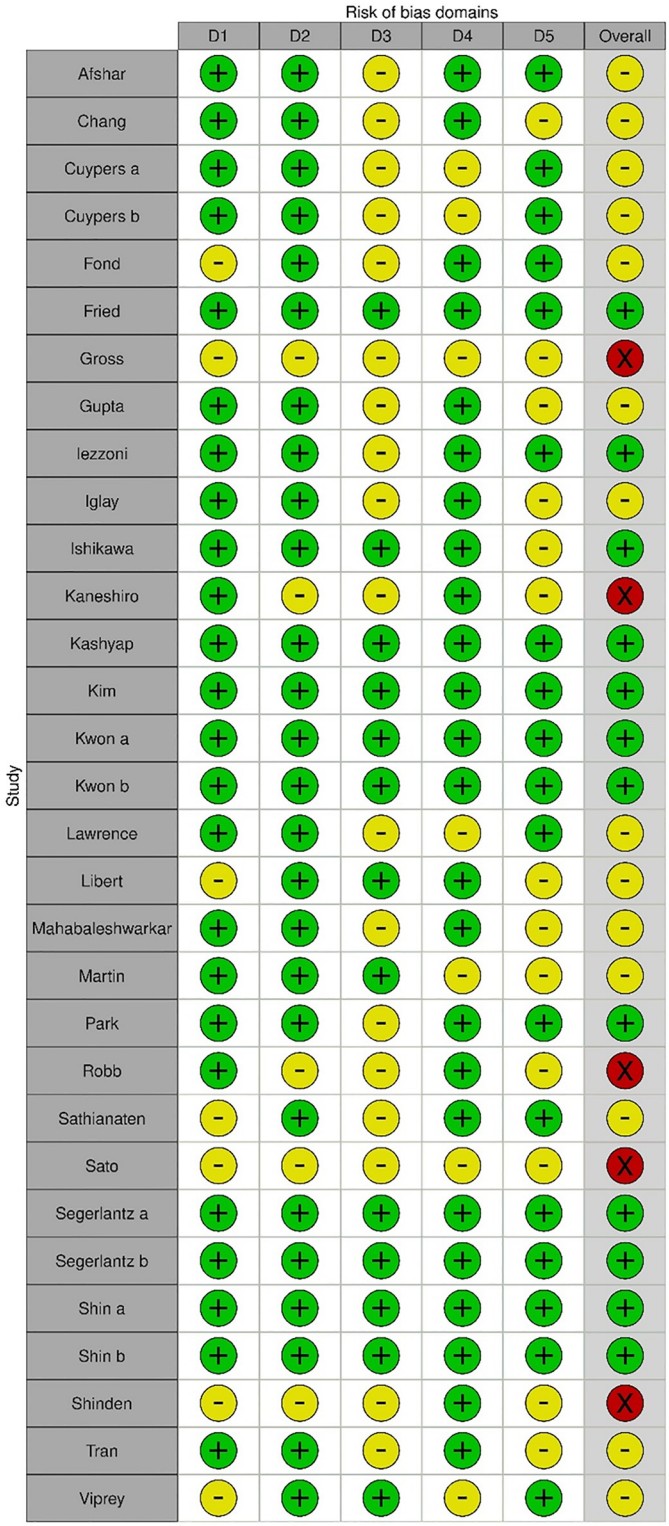

**Fig 2. Risk of bias.** D1 = Selection bias D2 = Information bias D3 = Misclassification bias; D4 = Confounding D5 = Missing data; Green = Low Yellow = Medium Red = High.

**Table 2. Outcomes of studies and type of disability.**

| AUTHOR | TYPE OF DISABILITY | PRIMARY OUTCOME | MEASURE IN PWD | MEASURE IN PEOPLE WITHOUT DISABILITIES | EFFECT MEASURE | SECONDARY OUTCOME | MEASURE IN PWD | MEASURE IN PEOPLE WITHOUT DISABILITIES | EFFECT MEASURE | TREND FOR PWD |
|---|---|---|---|---|---|---|---|---|---|---|
| Afshar [64] | Intellectual (learning disability) | 10-yr survival rate | 77.6% (95% CI = 72.2–83.3%) | 89.9% (95% CI = 89.4–90.3%) | 10-yr survival relative rate: 12.3% lower for PwD | 5-yr survival rate | 84% (95% CI = 79.9–88.4%) | 92.2% (95% CI = 91.8–92.5%) | 5-yr survival relative rate: 8.2% lower for PwD | WORSE |
| Chang [41] | Psychosocial (mental illness) | Access to state-of-the-art treatment | 68% received surgery | 82% received surgery | Adjusted OR of receiving surgery for PwD = 0.47 (95% CI = 0.34–0.65; P = 0.001) | 5-yr survival rate | 50.50% | 68.10% | Adjusted relative risk of death 1.58 higher for PwD (95% CI = 1.30–1.93; P,0.001). | WORSE |
| Cuypers [49] | Intellectual | Cancer-specific mortality | not mentioned | not mentioned | SMR = 1.48; (95% CI = 1.42–1.54) for PwD | n/a | n/a | n/a | n/a | WORSE |
| Cuypers [65] | Intellectual | Insurance claims for cancer hospital care | IR = 28.9 per 1000 person/year | IR = 45.3 per 1000 person/year | IRR = 0.64 (95% CI = 0.62–0.66) in PwD | n/a | n/a | n/a | n/a | WORSE |
| Fond [36] | Psychosocial (Severe psychiatric disease) | End-of-life treatment access | Incidence of palliative in month before death = 81.3% | Incidence of palliative in month before death = 75.2% | more trips to palliative care in last month of life (aOR 1.32, 95%CI [1.15–1.51], p<0.001) in last month of life in PwD | Overall survival time (days) | 886 | 918 | p value = 0.21 | NULL for mortality, WORSE for end-of-life treatment |
| Fried [44] | Psychosocial (Severe mental illness) | Cancer-specific 5-yr mortality | not mentioned | not mentioned | HR = 1.39 (95% CI: 1.04–1.84) for PwD | Access to state-of-the-art treatment | 12.8% received surgery | 21.8% received surgery | OR = 0.66 (95% CI: 0.49–0.89) for PwD of receiving surgery | WORSE |
| Gross [58] | Any | Screening results | not mentioned | not mentioned | PwD less often diagnosed for cancer through a mammography screening (OR for patients with physical impairment = 0.70; p < 0.05; OR for Sensory Impairment = 0.58; p < 0.05) than patients without disability. | Invasiveness of treatment | not mentioned | not mentioned | PwD less likely to receive breast conserving treatment (OR 0.58; p < 0.05) and more likely to have a mastectomy without reconstruction (OR = 1.96; p < 0.05) than those without disabilities | WORSE |
| Gupta [56] | Cognitive (Dementia) | Stage at diagnosis | 8.4% diagnosed on autopsy or death certificate | 1.9% diagnosed on autopsy or death certificate | aOR = 2.31 (95% CI 1.79–3.00) for PwD to have colon cancer reported only after death (i.e., from autopsy or death certificate) | Access to state-of-the-art treatment | not mentioned | not mentioned | aOR = 0.43 (95% CI 0.33–0.70) for PwD to receive surgery; aOR = 0.21 (95% CI 0.13–0.36) for PwD to to receive adjuvant chemo | WORSE |

*(Continued)*

**Table 2.** (Continued)

| AUTHOR | TYPE OF DISABILITY | PRIMARY OUTCOME | MEASURE IN PWD | MEASURE IN PEOPLE WITHOUT DISABILITIES | EFFECT MEASURE | SECONDARY OUTCOME | MEASURE IN PWD | MEASURE IN PEOPLE WITHOUT DISABILITIES | EFFECT MEASURE | TREND FOR PWD |
|---|---|---|---|---|---|---|---|---|---|---|
| Iezzoni [55] | Any | Cancer-specific mortality | not mentioned | not mentioned | HR = 1.37 (95% CI, 1.24–1.51) of cancer specific mortality for PwD | Access to state-of-the-art treatment | 68.5% received surgery | 82.2% received surgery | aRR 0.84 (95% CI 0.79–0.89) for PwD to receive surgery | WORSE |
| Iglay [37] | Psychosocial (Mental illness) | Treatment delay | 8.60% | 8.70% | aRR 1.36 (95% CI 1.06, 1.74) for PwD subgroup with severe mental illness of initial treatment delay at 60 days relative to controls | Diagnosis delay | 34.90% | 34.80% | aRR 1.11 (95% CI 1.00, 1.23) for PwD subgroup with comorbid anxiety and depression relative to controls | WORSE |
| Ishikawa [45] | Psychosocial (Schizophrenia) | Overall in-hospital mortality | 4.20% | 1.80% | OR = 1.35; (95% CI 1.04–1.75, P = 0.026) for pwd | Stage at diagnosis and access to state-of-the-art treatment | 33.9% stage IV; 56.5% surgery | 18.1% stage IV; 70.2% surgery | RR 1.86 (95% CI 1.72–2.00; P<0.001) of higher stage at diagnosis and OR = 0.77 (95% CI 0.69–0.85, P = 0.001) for access to surgical or endoscopic treatment for PwD | WORSE |
| Kaneshiro [46] | Psychosocial (Schizophrenia) | Incidence of invasive surgery | 84.3% mastectomy | 63.2% mastectomy | (P = 0.002) | Access to state-of-the-art treatment | 56% received radiotherapy | 75% received radiotherapy | (P = 0.078). | WORSE |
| Kashyap [47] | Psychosocial (Mental illness) | End of life Emergency Department use | 15.6% with access to ED in last 30 days of life | 13.3% with access to ED in last 30 days of life | p < 0.01 | Impact of outpatient mental health treatment in mental illness | not mentioned | not mentioned | aOR 0.82 (95% confidence interval 0.78–0.87) for mental health patients on outpatient mental health treatment to have multiple end-of-life ED visits | WORSE |
| Kim [22] | Any | Mortality | 125.2 per 1000 | 104.3 per 1000 | aHR = 1.18 (95% CI: 1.14–1.21) for PwD and aHR = 1.62 (95% CI: 1.56–1.69) for severe disability group | Mortality in patients who received surgery | not mentioned | not mentioned | aHR 1.21 (95% CI: 1.16–1.27), even higher in severe disability group (aHR 1.69, 95% CI: 1.57–1.81), | WORSE |
| Kwon [19] | Any | Median overall survival | 36.8 months | 51.2 months | p < 0.001 | Access to state-of-the-art treatment | 37.5% received transplant | 43.7% received transplant | p = 0.072 | WORSE |

(Continued)

**Table 2.** (Continued)

| AUTHOR | TYPE OF DISABILITY | PRIMARY OUTCOME | MEASURE IN PWD | MEASURE IN PEOPLE WITHOUT DISABILITIES | EFFECT MEASURE | SECONDARY OUTCOME | MEASURE IN PWD | MEASURE IN PEOPLE WITHOUT DISABILITIES | EFFECT MEASURE | TREND FOR PWD |
|---|---|---|---|---|---|---|---|---|---|---|
| Kwon [62] | Any | Median overall survival | 10.8 months | 17.1 months | p = 0.02 | Access to state-of-the-art treatment | chemo 71.2% vs 77.1%, P = .0031, and transplant 17.5% | chemo 77.1%; transplant 26.9% | p = 0.0031 and p = 0.002 | WORSE |
| Lawrence [40] | Psychosocial (Severe mental illness) | All-cause and cancer-specific mortality | not mentioned | not mentioned | all-cause mortality HR = 1.36; (95% CI 1.18, 1.57) and cancer-specific mortality HR = 1.21 (95% CI 1.03, 1.44) for women with SMI compared to controls | 10-year overall survival | 73.10% | 78.30% | not mentioned | WORSE |
| Libert [54] | Cognitive | Overall mortality | 12.3% at 2 years | 2% at 2 years | HR = 6.13 (95% CI = 2.07–18.09; p = 0.001) for people with cognitive impairment; HR = 3.06; (95% CI = 1.31–7.11, p = 0.009) for people with loss of instrumental autonomy | n/a | n/a | n/a | n/a | WORSE |
| Mahabaleshwarkar [43] | Psychosocial (mental illness) | Access to state-of-the-art treatment | not mentioned | not mentioned | aOR = 0.79 (95% CI = 0.65–0.97) of receiving guideline-consistent breast cancer treatment for PwD | Healthcare utilization | not mentioned | not mentioned | aIRR = 0.92 (95% CI = 0.89–0.94) for breast-cancer related outpatient visits; aIRR = 0.84 (95% CI = 0.71–0.99) for breast-cancer related ER visits for PwD | WORSE |
| Martin [57] | Cognitive | Overall mortality | not mentioned | not mentioned | HR 1.39 (95% CI = 1.09, 1.78, p>0.01) for PwD | Access to state-of-the-art treatment | 22.3% with mild, 35.6% with moderate and 51.8% with severe cognitive impairment received primary endocrine therapy (NOT state of the art) | 12.4% women with normal cognition received PET | p <0.001 | WORSE |

(Continued)

**Table 2.** (Continued)

| AUTHOR | TYPE OF DISABILITY | PRIMARY OUTCOME | MEASURE IN PWD | MEASURE IN PEOPLE WITHOUT DISABILITIES | EFFECT MEASURE | SECONDARY OUTCOME | MEASURE IN PWD | MEASURE IN PEOPLE WITHOUT DISABILITIES | EFFECT MEASURE | TREND FOR PWD |
|---|---|---|---|---|---|---|---|---|---|---|
| Park [61] | Any | Long-term all-cause mortality of 5-year cancer survivors | not mentioned | not mentioned | Male PwD HR = 1.48 (95% CI 1.33–1.66) and female PwD HR = 1.53 (95% CI, 1.28–1.83) compared with controls | Short-term (<5 years) all-cause mortality | not mentioned | not mentioned | male with impaired communication HR = 1.24 (95% CI, 1.07–1.44), female with internal organ disability HR, 2.20 (95% CI, 1.42–3.42) | WORSE |
| Robb [51] | Cognitive | Median overall survival | 23.0 months (0.2–140.7 months) | 72.6 months for controls (0.9–135.5 months) | p < 0.001 | n/a | n/a | n/a | n/a | WORSE |
| Sathianathen [42] | Psychosocial (Mental illness) | Access to state-of-the-art treatment | not mentioned | not mentioned | OR 0.55 (95% CI 0.37–0.81) for patients with severe mental illness and OR 0.71 (95%CI 0.58–0.88) for those with depression of receiving curative treatment. | Cancer-specific mortality | not mentioned | not mentioned | severe mental illness patients had HR 1.35 (95% CI1.14–1.61) in both the NMIBC (HR 1.48, 95% CI 115–1.92) and MIBC (HR1.37, 95% CI 1.10–1.72) subgroups, compared with controls | WORSE |
| Sato [59] | Any | Access to state-of-the-art treatment | not mentioned | not mentioned | difference not significant | n/a | n/a | n/a | n/a | NULL |
| Segerlantz [53] | Intellectual | Pain control prescription | 36% | 60% | RR 0.61 (95% CI 0.54–0.69) for PwD to have prescription of COX inhibitors, RR 0.63 (95% CI 0.53–0.73) for weak opioids | Prescription of other drugs | 36% on antidepressants; 47% on anxiolytics | 17% on antidepressants; 16% on anxiolytics | RR 2.09 (95% CI 1.74–2.51) for PwD to be prescribed antidepressants: RR 2.84 (2.39–3.38) for PwD to be prescribed anxiolytics | WORSE |
| Segerlantz [52] | Intellectual | Healthcare utilization | 1.5 visits per person in final year of life | 1.75 visits per person in final year of life | RR 0.90 (95% CI 0.87–0.93) for PwD to be less likely than controls to have >1 visit in specialist inpatient care during last year of life; | Quality of end-of-life care | 31% accessed advanced hospital care | 55% accessed advanced hospital care | RR 0.57 (95%CI 0.51–0.64) for PwD to have access to advanced hospital care | WORSE |

(Continued)

**Table 2.** (Continued)

| AUTHOR | TYPE OF DISABILITY | PRIMARY OUTCOME | MEASURE IN PWD | MEASURE IN PEOPLE WITHOUT DISABILITIES | EFFECT MEASURE | SECONDARY OUTCOME | MEASURE IN PWD | MEASURE IN PEOPLE WITHOUT DISABILITIES | EFFECT MEASURE | TREND FOR PWD |
|---|---|---|---|---|---|---|---|---|---|---|
| Shin [63] | Any | Overall mortality | 531.2 per 1000 | 463.1 per 1000 | aHR 1.08, (95% CI: 1.06–1.11) for PwD, and subgroup with severe disability HR = 1.20 (95% CI: 1.16–1.24) | Access to state-of-the-art treatment | 19.8% surgery; 42.3% chemo; 26.4% radiation | 21.9% surgery; 46.1 chemo; 27.6% radiation | aOR Surgery = 0.82, (95% CI 0.77–0.86), aOR chemo = 0.80, (95% CI: 0.77–0.84), aOR radiotherapy = 0.92 (95% CI: 0.88–0.96) for PwD | WORSE |
| Shin [60] | Any | Access to state-of-the-art treatment | Surgery 33.1%; ADT 57.9% | Surgery 38.6%; ADT 55% | Surgery aOR = 0.79, (95% CI 0.74–0.84); ADT aOR = 1.10 (95% CI1.04–1.16) for PwD. For severe disability, surgery aOR = 0.60 (95% CI, 0.54–0.67), ADT aOR = 1.29 (95% CI, 1.18–1.42) | Overall mortality and cancer-specific mortality | 57.3 per 1000; 26.7 per 1000 | 43.7 per 1000; 21.7 per 1000 | Overall mortality aHR, 1.20 (95% CI, 1.15-1-25) for PwD; with severe disability aHR 1.47 (95% CI 1.37–1.57). Cancer-specific mortality aHR 1.11 for pwd (1.04–1.18), but no difference when PwD had same access to surgery. | WORSE |
| Shinden [38] | Psychosocial (Mental illness) | Access to state-of-the-art treatment | total mastectomy 78%, postoperative adjuvant chemo 0%, radiation 2% | total mastectomy: 59%; postoperative adjuvant chemo 19%; radiation 18% | p <0.05 for all the mentioned outcomes | Overall survival | not mentioned | not mentioned | no difference | NULL for mortality, WORSE for treatment |
| Tran [39] | Psychosocial (Schizophrenia) | Overall mortality and All-cancer-mortality | not mentioned | not mentioned | 4-fold higher all-cause mortality for schizophrenia. Cancer SMR = 1.5 (95% CI: 1.2–1.9). | Mortality by cancer type | not mentioned | not mentioned | Male PwD and lung SMR = 2.2 (95% CI, 1.6–3.3); female PwD and breast SMR = 2.8 (95% CI, 1.6–4.9) compared to controls | WORSE |

*(Continued)*

**Table 2.** (Continued)

| AUTHOR | TYPE OF DISABILITY | PRIMARY OUTCOME | MEASURE IN PWD | MEASURE IN PEOPLE WITHOUT DISABILITIES | EFFECT MEASURE | SECONDARY OUTCOME | MEASURE IN PWD | MEASURE IN PEOPLE WITHOUT DISABILITIES | EFFECT MEASURE | TREND FOR PWD |
|---|---|---|---|---|---|---|---|---|---|---|
| Viprey [48] | Psychosocial (Schizophrenia) | Access to state-of-the-art treatment | early palliative care: 77%; end-of-life chemo: 10%; end of life surgery: 17% | early palliative care: 72%; end-of-life chemo: 15%; end of life surgery: 20% | aOR for early palliative care = 1.27 (95% CI = 1.03;1.56; p = 0.04), aOR for end-of-life chemo = 0.53 (0.41–0.70, p<0.0001), aOR end-of-life surgery = 0.73 (0.59;0.90, p<0.01) for PwD. | Quality of end-of-life care | Hospitalization in acute care unit the month before death 33%; median length of last hospital stay 13 days; deaths in the ICU/ED 10% | Hospitalization in acute care unit the month before death 24%; median length of last hospital stay 10 days; deaths in the ICU/ED 11% | aOR for hospitalization in acute care unit the month before death = 1.41 (95% CI = 1.18;1.67; p<0.001); longer length of last hospital stay (Beta = 1.22, SD = 0.05; p<0.0001); aOR for deaths in the ICU/ED = 0.74 (95% CI = 0.56;0.97; p = 0.04) for PwD. | WORSE |

LEGEND OF ABBREVIATIONS for Table 2:

yr = year; CI = Confidence Interval; OR = Odds Ratio; SMR = Standardized Mortality Ratio; IR = Incidence Rate; IRR = Incidence Rate Ratio; aOR = adjusted Odds Ratio; HR = Hazard Ratio; PWD = PwD; aRR = adjusted Risk Ratio; ED = Emergency Department; aHR = adjusted Hazard Ratio; SMI = Severe Mental Illness; aIRR = adjusted Incidence Rate Ratio; PET = Primary Endocrine Therapy; NMIBC = non-muscle invasive bladder cancer; MIBC = muscle invasive bladder cancer; COX = cyclooxygenase; RR = Relative Risk; ADT = Androgen Deprivation Therapy; ICU = Intensive Care Unit

intellectual disability were more likely than controls to have worse access to specialist care in the last year of life [52].

Regarding other factors contributing to worse outcomes, one study showed an association between worse access to screening programs (for breast, colon and cervix cancers) and higher cancer specific mortality, underlining the inequality in screening practices [50]. Two other studies highlighted an association between barriers to screening and worse outcomes for gastric and breast cancer in PwD [22, 46]. Two studies detected even worse disparities in access to state-of-the-art treatment or end-of-life care related to ethnicity and age, with young disabled non-white men having the worst outcomes [47, 55]. One study of people with intellectual disabilities with any type of cancer suggested worse underdiagnosis for older females, while another focusing on breast cancer detected an association between physical disability and inappropriate invasiveness of treatment [49, 58]. In a study of non-small cell lung cancer patients, those with respiratory or nervous system disabilities had the lowest chance of receiving guideline-appropriate surgery, while another paper on lung cancer recorded the worst access to treatment among people with communication or neurologic disabilities [55, 63]. A study about stomach cancer and patients with all kinds of disabilities also found an association between worse outcomes and severe intellectual impairment [22].

Finally, three studies showed an association between worse outcomes of treatment access and poverty among people with cancer [19, 48, 63].

## Discussion

In our review, compared to people without disabilities, PwD were found to have worse survival, higher overall and cancer-specific mortality, loss of chance for access to state-of-the-art care or curative-intent therapies, treatment delays, undertreatment or excessively invasive treatment, worse access to in-hospital services, less specialist healthcare utilization, more difficult access to pain medications and inadequate end-of-life quality of care. Only one eligible study found no difference in overall survival or disease-free survival between patients with and without disabilities; it was a small paper with a high risk of bias, with a cohort including only operable breast cancer in a small number of patients, and it still showed an association between disability and excessively invasive breast surgery without any clear cancer-related clinical reason. Furthermore, the incidence of disability in its retrospective cohort was inexplicably only half of the known national incidence, suggesting severe misclassification bias [38]. These finding suggest that differences in frequency of appropriate treatment appear to explain the higher cancer-specific mortality for this vulnerable population, with higher mortality likely due to loss of chance and unequal clinical care. Even if sometimes treatment decisions for PwD can be clinically complex, such as the above-mentioned case of cognitive impairment with legal consent or non-compliance issues, or when confronted with a disability-related shortened life expectancy or frailty for some syndromes, there is no plausible medical justification for such a wide disparity compared to patients without disabilities, and these results raise severe concerns about equality in cancer care [67, 68].

The results of this study are consistent with those of other recent literature reviews, showing that PwD experience inequities at several points throughout the cancer care pathway [12, 24, 68, 69]. Screening disparities have been known and documented for years: they can vary by disability type, severity, healthcare offer and social or demographic situations, with some differences across countries, but globally there is a largely similar trend of major barriers to screening for people with disability, showing a clear need to improve the inclusiveness of these early-diagnosis services [12, 24, 68].

Providing equitable cancer care has to start early in the cancer clinical pathway, because delays in receiving a diagnosis tend to lead to late access to treatment and worse outcomes [67]. A frequently observed issue is that new signs and symptoms tend to be attributed to often to the underlying disability, a clinical mistake called "diagnostic overshadowing" [70]. A recent scoping review about cancer outcomes in adults with intellectual and developmental disabilities has described disparities at every step of the way, from screening, to staging, to treatment and survival outcomes, recognizing how these experiences do not originate simply from a gap in early diagnosis, but from larger structural issues that ultimately hinders quality of the entire cancer care pathway [25]. Another review of cancer treatments for people with intellectual disabilities highlighted possible themes that might interfere with treatment, such as genetic syndrome frailty that might render certain drug treatments too dangerous, the issue of behavioral non-compliance in a subgroup of patients, and problems related to legal capacity and obtainment of informed consent. Still, these three factors should not represent an insurmountable barrier because with appropriate arrangements (e.g. pharmacology consults, procedural sedation, legal assistance) it should still be possible to offer guideline-consistent treatment to patients [68]. With physical disabilities clinical decisions can sometimes be objectively more difficult than in people without disabilities, because of concerns about baseline performance status or competing health risks due to invasive or toxic treatments that might result in further dramatic loss of function (e.g. possible loss of postoperative upper limb function after breast surgery in patients with previous spinal cord injury and lower limb paralysis); this has been described as a compelling reason to move towards better cooperation between cancer care clinicians and disability specialists who have been in charge of the patient well before their oncology episode, and also as one of the fundamental facts that make shared decision making with patients (or sometimes their families or caregivers) of the utmost importance [12].

This clear evidence of inequities emphasizes the very urgent need for better cancer care for PwD. Furthermore, disparity in healthcare for PwD is not unique to oncology, as research about the recent Covid-19 pandemic has clearly proven [71–76]. During the pandemic, PwD have died in disproportionate numbers–almost three times as much globally than people without disabilities—and have been excluded from the decision-making process, because their needs have been ignored; they have ended up facing an increasing amount of psychological distress, lack of social support, extreme isolation, food insecurity, disparities in health care access and even discrimination at work. In many cases, government response has compromised the human rights of disabled people, having exposed and magnified existing structural failings and inequalities [74, 75, 77, 78].

Recently, the second report of the Missing Billion Initiative has called for reimagining health systems with a vision of inclusive health informed by diverse perspectives of PwD, who are still facing worse health outcomes across SDG3 indicators (Sustainable Development Goal 3 by the WHO, i.e. ensure healthy lives and promote well-being for all at all ages), globally and with all sort of impairment types [78]. The first Missing Billion report had highlighted a substantial life expectancy gap of 10–20 years for PwD, with all-cause mortality rates approximately twice as high as those of people without disabilities [76]. Closing this gap is now a priority, but it requires long-term investments to design from the start health systems that expect, accept and connect PwD, with sufficient earmarked funds, dedicated leadership and clear governance based on data and evidence disaggregated by disability. Ideally service delivery should aim at affordability, autonomy of patients, accessibility, specific workforce skillsets and availability of rehabilitation services [78]. The multiple barriers experienced by PwD during their cancer care (Fig 3) are hence a reflection of a broader process of discrimination and disadvantage, mirrored in structural failings of current healthcare systems, within networks of intersecting factors that ultimately influence cancer outcomes [25, 79, 80].

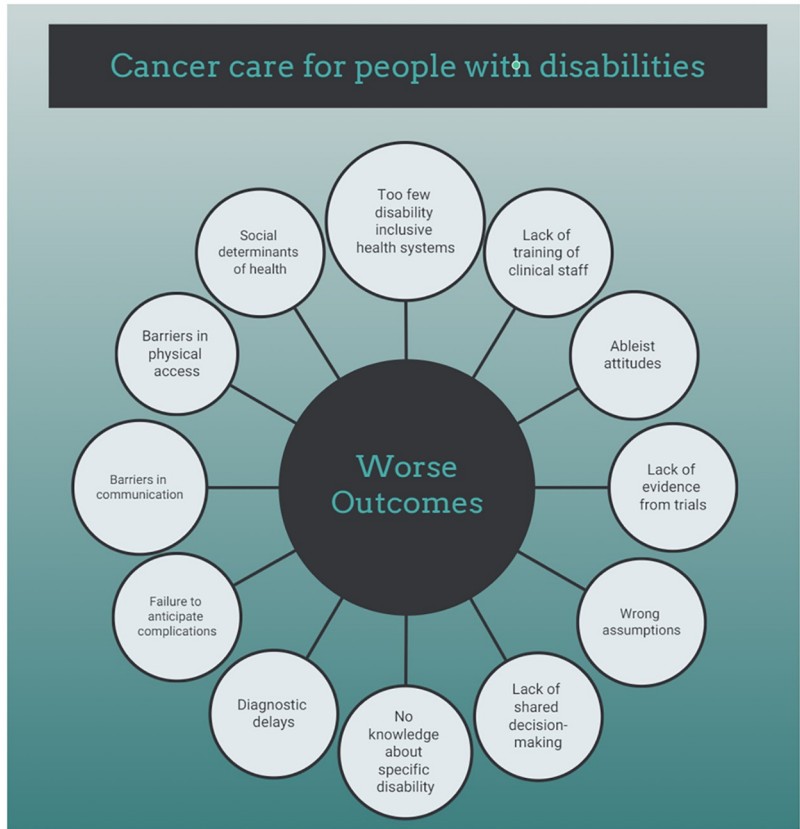

**Fig 3. Barriers experienced by PwD during cancer care.**

Healthcare workers need to receive evidence-based and appropriate training about disabilities, directly involving PwD and using a monitoring system to measure cultural progress and outcome improvement. This could help foster a change to move away from the ableist attitudes that are too often still observed, contributing to wrong assumptions and subsequent mistakes of diagnostic overshadowing or failure to anticipate specific complications [12, 81]. In a very recent qualitative study from the USA, interviewed physicians, mostly middle-aged white males, felt inadequately reimbursed for accommodations required by the 1990 Americans with Disabilities Act; according to some of these doctors, these concerns simply led them to discharge patients with disabilities [82]. Many physicians openly spoke about the lack of accessibility in their clinics without any plans to improve it, and several demonstrated a complete lack of disability knowledge about how to manage very basic issues–even stating that they were regularly sending patients to a zoo, cattle processing plant or supermarket to obtain a weight if a patient was in a wheelchair and unable to stand; several doctors admitted that they rarely spoke to these patients, regardless of the patients' ability to communicate, and that information was almost exclusively obtained from the caregiver. This confirms findings from previous qualitative research, that had described a lack of skills by healthcare workers to feel empathy for the embodied experience of living with a disability, with an obstinate resistance to adapting their habitual practice to these patients [10].

The importance of inclusive clinical trials to close the evidence gap about what works to improve cancer care for PwD cannot be overemphasized [79, 82–84]. There is still too little

evidence about how to treat cancer in the population with disabilities, which is very diverse and can have widely different therapeutic needs (hence existing services must be offered in a flexible, respectful, inclusive and accessible way to be relevant for this patient population). Thus, it is of the utmost importance to include PwD in clinical trials in oncology—both for curative-intent interventions and for palliative treatments; yet, historically they have been left out of studies, due to many factors such as ableist prejudice, or multiple barriers such as accessibility of research facilities and access to transportation, or lack of caregivers' engagement [83, 84]. Unfortunately, in clinical trials pre-existing conditions are often excluded, even if the conditions have little bearing on the treatment being tested or the outcome of the trial. Excluding disabled individuals from a study can result in a study population that does not even represent the general population, since disability often correlates with other inequalities (such as poverty and unemployment). The importance of targeting the recruitment of disabled individuals into clinical trials, as well as considering the unique barriers and motivations of this population, needs to be highlighted. A person with a disability may have difficulty traveling to a trial site; moreover, healthcare organizations should consider their audiences' digital literacy and the accessibility of their communications. Funds should be allocated to improve healthcare communication, adapting multiple formats, using captions and alt-text or pictorial representations of concepts as appropriate for the specific context. In addition, disabled individuals appear to be underrepresented as investigators in scientific research [85]. Despite 19% of the UK's general population identifying as disabled, only 4% of academic, research, and teaching staff do. Even if 25% of American adults live with a disability, in 2020 only 4% of US STEM PhDs were awarded to people with impaired hearing or vision, and just 1% to people with a mobility limitation. More disability-confident schemes and unconscious bias training could at least partially mitigate hiring discrimination, creating an academic workforce that better reflects the community in which it is based [86]. Recent evidence-based recommendations to promote inclusion in clinical trials include improving culture and sensitivity of staff through continuous education, receiving ongoing feedback from a community advisory panel during studies and increasing staff diversity to make sure underprivileged groups are represented [87].

Physicians and PwD should be able to collaborate along care pathways with shared-decision making, an approach based not only on clinical technical advice but on the life experience of patients, their caregivers and families, according to the principle of "Nothing About Us Without Us" [12]. In the clinical setting, barriers in physical access should be removed to avoid unacceptable delays in diagnosis and treatment [79, 80, 88]. Barriers in communication should be eliminated at several levels, from overcoming communication obstacles (not only for the hearing or visually impaired patients, but also with special-needs assistance for intellectual disability), to improving education of patients, clinicians and caregivers about cancer and the importance of screening, to training healthcare workers about the emotional and physical needs of PwD, enhancing cooperation with other specialists caring for them, in cross-functional teams, to anticipate and possibly reduce the impact of complications, with the goal of a patient-centred pathway [89]. Good communication is the foundation of achieving quality patient-centered care: assumptions about preferences can pose a risk like inaccurate information leading to medical errors and misdiagnoses. A recent qualitative study in the USA has shown that, in spite of healthcare workers trying their best, there are still many unsolved issues at this level and even many situations where physicians' preferences go against patients' wishes [87].

The strengths of this study include having followed PRISMA and ICM50 guidelines for systematic reviews; furthermore, the search strategy was based on a list of proven disability-specific terms and applied to the five largest medical databases analysing a twenty-year span of publications. This work has several limitations: firstly, the search strategy, limited to five

databases and to English language only, might not be fully comprehensive; we did not include studies published in non-English languages because of resource challenges with respect to costs, time, and expertise in non-English languages, but their inclusion would have likely increased generalizability and reduced the overall risk of bias. Furthermore all the eligible papers were from high-income countries, limiting the generalizability of the results, even if there is no reason why the situation should be very different in low and middle-income countries., Qualitative papers and grey literature were not included in the search strategy, hence the views and opinions of PwD about their cancer care were not investigated. Study selection was performed by only one reviewer, which implies a lack of independent screening. We also did not perform tests to measure publication bias due to the high heterogeneity of the eligible studies; although methods exist for simultaneous assessment of heterogeneity and publication bias, and potential differential publication bias, they require very large meta-analysis to reliably disentangle their effects [35]. Moreover, only one reviewer evaluated papers for risk of bias. Finally, the findings were very diverse hence it was not possible to conduct a meta-analysis, and approximately 15% of the studies had a high risk of bias Almost half (45%) of the eligible studies had a medium risk of bias, mostly due to possible misclassification bias for inclusion of PwD based on disability records (that have a tendency to miss mild cases) or missing data like details about cancer treatment goals, behavioral factors or date of diagnosis [36, 37, 42, 43, 49, 54, 56, 64, 65]. Approximately 15% of the studies had a high risk of bias due to factors such as having a very small sample size, a short follow-up, low data quality, a biased cohort or using a self-reporting survey [38, 46, 51, 58, 59]. There are still gaps in knowledge about quality of cancer care for people with disability that remain unanswered based on our findings, such as whether certain subgroups of disabilities or cancer types experience more significant disparities, or how other social determinants of health might come into play (as many PwD are caught in a cycle of poverty and deprivation); more data is needed on these topics to allow disaggregated analyses. Further research is also needed to evaluate the effectiveness of specific training of healthcare workers on quality of care for these patients.

In conclusion, PwD often experience severe disparities in cancer care compared to people without disabilities; physical and cultural barriers at different levels must be eliminated to ensure they receive equitable care. There is an urgent need for a robust health policy effort by governments, reimagining health systems with a vision of inclusive health and a sustained commitment, building on decades of progress on disability rights and engaging the participation of PwD at all levels.

## Supporting information

**S1 File. Supplement 1.** Search strategy; criteria for determining risk of bias; S1 Table with details of study characteristics.
(DOCX)

## Acknowledgments

I.T. would like to thank Danae Rodriguez Gatta, for assistance with disability search terms, and Dr. Meena Cherian, for many fruitful discussions about health systems and development goals.

## Author Contributions

**Conceptualization:** Hannah Kuper.

**Data curation:** Irene Tosetti.

**Formal analysis:** Irene Tosetti.

**Investigation:** Irene Tosetti.

**Methodology:** Hannah Kuper.

**Project administration:** Irene Tosetti.

**Supervision:** Hannah Kuper.

**Validation:** Hannah Kuper.

**Visualization:** Irene Tosetti.

**Writing – original draft:** Irene Tosetti.

**Writing – review & editing:** Hannah Kuper.

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
