## [Decision Letter · Decision Letter 0]

20 Jun 2023

PONE-D-23-10494Do people with disabilities experience disparities in cancer care? A systematic reviewPLOS ONE

Dear Dr. Tosetti,

Thank you for submitting your manuscript to PLOS ONE. After careful consideration, we feel that it has merit but does not fully meet PLOS ONE’s publication criteria as it currently stands. Therefore, we invite you to submit a revised version of the manuscript that addresses the points raised during the review process.Invited reviewers have provided several beneficial comments on this submission which I think could be implemented to improve the prepared material before any decision for publication.

We look forward to receiving your revised manuscript.

Kind regards,

Sina Azadnajafabad, MD, MPH

Academic Editor

PLOS ONE

Reviewers' comments:

Reviewer's Responses to Questions

**Comments to the Author**

1. Is the manuscript technically sound, and do the data support the conclusions?

Reviewer #1: Yes

Reviewer #2: Yes

2. Has the statistical analysis been performed appropriately and rigorously? 

Reviewer #1: N/A

Reviewer #2: N/A

3. Have the authors made all data underlying the findings in their manuscript fully available?

Reviewer #1: Yes

Reviewer #2: Yes

4. Is the manuscript presented in an intelligible fashion and written in standard English?

Reviewer #1: Yes

Reviewer #2: Yes

5. Review Comments to the Author

Reviewer #1: General Comments:

This study addresses an interesting topic. It includes a valuable effort in answering the questions about the disparities about treatment outcomes of a vulnerable population of patients. Overall, the abstract effectively communicates the objectives, methods, findings, and limitations of the systematic review. It highlights the need for further research and action to address the disparities in cancer care for people with disabilities. It is well-written and provides valuable insights into an important and underexplored area of healthcare. However, I think a few concerns need to be addressed in the next revision:

Study selection:

- Line 123, Lack of independent screening: The manuscript mentions that one author screened the studies by title, abstract, and full text to determine eligibility. Ideally, study selection should involve at least two independent reviewers who screen the studies separately and then compare their results to resolve any discrepancies. This helps minimize the potential for bias or errors in the selection process.

- Line 115: The exclusion criteria in the PRISMA flowchart (after retrieval) should match the exclusion criteria in the methods section. As I read, the exclusion in the methods were study type, study measure, study population and study outcome but not study setting. You should fully mention the actual exclusion (As shown by PRISMA chart) in the methods section.

Risk of Bias:

- Line 231: The authors should avoid speculation about the results in this section ( “…mostly due to possible misclassification bias…”). The justification for the medium risk of bias should be completely transferred to the discussion section followed by explanation and citations from the appropriate source.

Outcome results:

- I suggest starting this section with the main results of outcomes (line 249) and then mentioning the opposing result from the other study (small study).

- Line 246: The opposing result about one study has been reported by the authors, but the explanation should be omitted from here and be transferred to discussion.

- Line 253: This line includes an important claim about the association of disabilities with the worse outcomes in cancer patients. However, the reader needs to see the reported numbers from three cited articles that support this claim. Moreover, this claim is missing an important assumption: the heterogeneity of outcome measures in the reported articles. Which is only assessable via visual, statistical assessment and subgroup analysis of the study outcomes.

- Line 255: “…with a large portion of that disparity not justifiable on the basis of rational clinical decisions”. I do not find this claim to be true based on the findings of this manuscript. Please add extra details here to support this claim.

- Line 263: It would be great to limit the outcome reports about the opposing study to actual findings of the study and consider moving the suggestions about these findings to the discussion section.

- Same for lines 269, 289.

Discussion:

The discussion section is comprehensive, Including all the necessary concerns about the findings of the study. However, few points need to be addressed:

- Line 318: This line includes a very long sentence, which is hard to read. Please consider dividing it to smaller sections.

- The limitation about lack of independent screening should be addressed in the discussion or the screening needs to be performed by another person independently.

Figure 1:

- I suggest omitting the word “Wrong” in the excluded reports, since it might lead to misconception about the content of the excluded studies.

Table 2:

A few points to consider in the table 2 are:

- The overall quality of the table is acceptable; however, I suggest that all the abbreviations in the table be fully mentioned right below the table to make them understandable for the readers.

Some of these abbreviations include: IR , OR, PWD, aHR IRR, SMR and etc.

- I suggest specifying the type/s of disability to this table and changing the title accordingly. This is additional to the data extraction table (Table 1 Supplement1).

-

Some proofing notes to consider:

- Line 216: All the abbreviations in the manuscript must be fully mentioned the first time they appear in the text.

- Line 75 vs line 77: The citations in the manuscript should follow the same pattern through the manuscript (before or after the “.”) . Please apply the same pattern for all citations in the text according to previously published articles in PLOS One.

- Table 2 : lezzoni study : “disabilities”, Kaneshiro study : “Invasiveness”, Shinden study : “Postoperativ-e” + change “postop” to postoperative.

- Line 422 : “enroll”

Thank you for submitting this article, I hope that the suggestions can improve the overall quality of your valuable research.

Reviewer #2: This systematic review aimed to bridge an existing gap in the literature by conducting a comprehensive analysis of studies spanning a period of over two decades, addressing issues ranging from inadequate training of healthcare workers to the necessity for more inclusive clinical trials. Nevertheless, there are minor aspects of the study that necessitate additional clarification and improvement before considering the manuscript for publication. I have outlined these points in the comments provided below:

1. Abstract: The abstract provides a concise summary of the systematic review article. It effectively presents the background, methods, results, and discussion. However, I suggest including more specific details on the inclusion and exclusion criteria applied in the systematic review to enhance transparency and replicability. Additionally, it would be beneficial to briefly discuss the implications of the findings and potential strategies for addressing the identified disparities in cancer care for people with disabilities.

2. Introduction: The introduction provides a broad overview of the topic, highlighting the burden of cancer, the importance of accurate diagnosis and access to appropriate treatment, and the disparities faced by people with disabilities. I suggest providing some statistics to support the statements regarding the disparities in the use of cancer prevention services and lower cancer screening rates as factors contributing to higher mortality rates in people with disabilities. Also, consider emphasizing the potential impact of addressing healthcare disparities for people with disabilities on overall health outcomes and mortality rates.

3. Methods: The methods section provides a detailed systematic review process. However, consider discussing any efforts made to minimize publication bias. Additionally, it would be beneficial for the authors to provide a brief rationale for excluding non-English studies. Regarding the risk of bias assessment, report how many reviewers assessed the risk of bias in each study.

4. Results: The results section provides a comprehensive overview of the findings from the systematic review. It covers the study selection process, characteristics of included studies, types of disabilities and cancer, outcomes assessed, risk of bias assessment, and specific results for each outcome. It would be helpful to discuss if there were any patterns or differences in outcomes among the different types of disabilities or cancers. Are there certain subgroups within disabilities or cancer types that experience more significant disparities in cancer care? What about the clinical significance of the findings? How do these disparities in cancer care for people with disabilities impact patient outcomes, quality of life, and healthcare delivery?

5. Discussion: I suggest that the authors explicitly mention the gaps in knowledge or research questions that remain unanswered based on the findings of this study. Also, consider discussing strategies for promoting the inclusion of people with disabilities in clinical trials and highlight the challenges that exist in communication and partnership between healthcare providers and individuals with disabilities.

Overall, this systematic review article appears to be well-structured, with a comprehensive search strategy, a clear presentation of results, and a thoughtful discussion of limitations. The findings have significant implications for healthcare policies and practices, particularly in addressing the disparities in cancer care for individuals with disabilities. I would recommend that the authors continue with the manuscript submission process while addressing the minor comments and suggestions for improvement mentioned above.

6. PLOS authors have the option to publish the peer review history of their article (what does this mean?). If published, this will include your full peer review and any attached files.

Reviewer #1: **Yes: **Amirhossein Parsaei

Reviewer #2: **Yes: **Parinaz Paranjkhoo, MD, MPH

---

## [Author Response · Author response to Decision Letter 0]

30 Jul 2023

REVIEWER 1 COMMENTS:

Study selection:

- Line 123, Lack of independent screening: The manuscript mentions that one author screened the studies by title, abstract, and full text to determine eligibility. Ideally, study selection should involve at least two independent reviewers who screen the studies separately and then compare their results to resolve any discrepancies. This helps minimize the potential for bias or errors in the selection process.

Thank you for this comment. The lack of independent screening has been added in the abstract and discussion sections as a limitation of our study.

- Line 115: The exclusion criteria in the PRISMA flowchart (after retrieval) should match the exclusion criteria in the methods section. As I read, the exclusion in the methods were study type, study measure, study population and study outcome but not study setting. You should fully mention the actual exclusion (As shown by PRISMA chart) in the methods section.

Thank you for pointing this out; the paragraph has now been updated to match the PRISMA flowchart.

Risk of Bias:

- Line 231: The authors should avoid speculation about the results in this section ( “…mostly due to possible misclassification bias…”). The justification for the medium risk of bias should be completely transferred to the discussion section followed by explanation and citations from the appropriate source.

This part has now been removed from results and moved to discussion, as kindly suggested.

Outcome results:

- I suggest starting this section with the main results of outcomes (line 249) and then mentioning the opposing result from the other study (small study).

The beginning of this section has been edited according to this recommendation.

- Line 246: The opposing result about one study has been reported by the authors, but the explanation should be omitted from here and be transferred to discussion.

We appreciate your suggestion and have now moved this commentary to the discussion section.

- Line 253: This line includes an important claim about the association of disabilities with the worse outcomes in cancer patients. However, the reader needs to see the reported numbers from three cited articles that support this claim. Moreover, this claim is missing an important assumption: the heterogeneity of outcome measures in the reported articles. Which is only assessable via visual, statistical assessment and subgroup analysis of the study outcomes.

Thank you, we have added the reported numbers from several articles. We have added comments about heterogeneity in the discussion section. 

- Line 255: “…with a large portion of that disparity not justifiable on the basis of rational clinical decisions”. I do not find this claim to be true based on the findings of this manuscript. Please add extra details here to support this claim.

We appreciate your remark; we have now added more data about these differences in treatment access for people with disabilities.

- Line 263: It would be great to limit the outcome reports about the opposing study to actual findings of the study and consider moving the suggestions about these findings to the discussion section.

- Same for lines 269, 289.

Thank you, the comments about findings have all been moved to the discussion section. 

Discussion:

The discussion section is comprehensive, Including all the necessary concerns about the findings of the study. However, few points need to be addressed:

- Line 318: This line includes a very long sentence, which is hard to read. Please consider dividing it to smaller sections.

We have improved the readability of the whole section; thank you for this remark.

- The limitation about lack of independent screening should be addressed in the discussion or the screening needs to be performed by another person independently.

Thank you for pointing out this important limitation. We have now added the lack of independent screening to the list of study limitations, both in Discussion and Abstract.

Figure 1:

- I suggest omitting the word “Wrong” in the excluded reports, since it might lead to misconception about the content of the excluded studies.

We have omitted that word and used the term “ineligible”.

Table 2:

A few points to consider in the table 2 are:

- The overall quality of the table is acceptable; however, I suggest that all the abbreviations in the table be fully mentioned right below the table to make them understandable for the readers.

Some of these abbreviations include: IR , OR, PWD, aHR IRR, SMR and etc.

We have updated the table with a legend of abbreviations. 

- I suggest specifying the type/s of disability to this table and changing the title accordingly. This is additional to the data extraction table (Table 1 Supplement1).

The type of disability has been added to a new column in Table 2 and the title has been updated accordingly.

Some proofing notes to consider:

- Line 216: All the abbreviations in the manuscript must be fully mentioned the first time they appear in the text.

- Line 75 vs line 77: The citations in the manuscript should follow the same pattern through the manuscript (before or after the “.”) . Please apply the same pattern for all citations in the text according to previously published articles in PLOS One.

- Table 2 : lezzoni study : “disabilities”, Kaneshiro study : “Invasiveness”, Shinden study : “Postoperativ-e” + change “postop” to postoperative.

- Line 422 : “enroll”

Thank you for submitting this article, I hope that the suggestions can improve the overall quality of your valuable research.

Thank you for mentioning these proofreading mistakes, they have been corrected.

REVIEWER 2 COMMENTS:

This systematic review aimed to bridge an existing gap in the literature by conducting a comprehensive analysis of studies spanning a period of over two decades, addressing issues ranging from inadequate training of healthcare workers to the necessity for more inclusive clinical trials. Nevertheless, there are minor aspects of the study that necessitate additional clarification and improvement before considering the manuscript for publication. I have outlined these points in the comments provided below:

1. Abstract: The abstract provides a concise summary of the systematic review article. It effectively presents the background, methods, results, and discussion. However, I suggest including more specific details on the inclusion and exclusion criteria applied in the systematic review to enhance transparency and replicability. Additionally, it would be beneficial to briefly discuss the implications of the findings and potential strategies for addressing the identified disparities in cancer care for people with disabilities.

Thank you for your comments. We have added more information about inclusion and exclusion criteria as kindly requested, as well as implications of the findings and potential strategies. 

2. Introduction: The introduction provides a broad overview of the topic, highlighting the burden of cancer, the importance of accurate diagnosis and access to appropriate treatment, and the disparities faced by people with disabilities. I suggest providing some statistics to support the statements regarding the disparities in the use of cancer prevention services and lower cancer screening rates as factors contributing to higher mortality rates in people with disabilities. Also, consider emphasizing the potential impact of addressing healthcare disparities for people with disabilities on overall health outcomes and mortality rates.

We have added statistics to support the statements regarding disparities in the use of cancer prevention services in people with disabilities. We also added a paragraph about the clinical importance of addressing these disparities.

3. Methods: The methods section provides a detailed systematic review process. However, consider discussing any efforts made to minimize publication bias. Additionally, it would be beneficial for the authors to provide a brief rationale for excluding non-English studies. Regarding the risk of bias assessment, report how many reviewers assessed the risk of bias in each study.

We chose not perform tests to measure publication bias – and it is a limitation of our study - due to the high heterogeneity of the eligible studies; although methods exist for simultaneous assessment of heterogeneity and publication bias, and potential differential publication bias, they require very large meta-analysis to reliably disentangle their effects(1). We have now mentioned the lack of publication bias assessment both in the Abstract Methods and Discussion sections, explaining that this is a limitation of our study.

We did not include studies published in non-English languages because of resource challenges with respect to costs, time, and expertise in non-English languages; however, inclusion would have likely increased generalizability and reduced the overall risk of bias. Results of studies examining the impact of including non-English trials on review estimates of effect have been mixed, with some showing no difference while some others concluding the opposite. (2–5) We have further described the reasons for our choice and how we consider a limitation in the Discussion section.

Only one reviewer evaluated risk of bias for the studies in each study, and we now mention this in Methods and Discussion as a limitation.

4. Results: The results section provides a comprehensive overview of the findings from the systematic review. It covers the study selection process, characteristics of included studies, types of disabilities and cancer, outcomes assessed, risk of bias assessment, and specific results for each outcome. It would be helpful to discuss if there were any patterns or differences in outcomes among the different types of disabilities or cancers. Are there certain subgroups within disabilities or cancer types that experience more significant disparities in cancer care? What about the clinical significance of the findings? How do these disparities in cancer care for people with disabilities impact patient outcomes, quality of life, and healthcare delivery?

There was insufficient data to allow identification of subgroups of disability or cancer types that experienced more significant disparities in cancer care, and more data is needed on this topic to allow these disaggregated analyses. As for the clinical impact of disparities, we narratively described an association between disability and higher overall and cancer-specific mortality, an association with loss of chance of receiving guideline-consistent cancer treatment and with worse end-of-life care. More details are in the “outcome results” section. 

5. Discussion: I suggest that the authors explicitly mention the gaps in knowledge or research questions that remain unanswered based on the findings of this study. Also, consider discussing strategies for promoting the inclusion of people with disabilities in clinical trials and highlight the challenges that exist in communication and partnership between healthcare providers and individuals with disabilities.

We have added a paragraph about recommendations on practical measures to improve inclusion of people with disabilities in clinical trials, and also expanded our comments on specific issues in communication between doctors and disabled patients according to current literature. Furthermore we have expanded the part of discussion about the knowledge gaps that remain unanswered.

---

## [Decision Letter · Decision Letter 1]

24 Aug 2023

PONE-D-23-10494R1Do people with disabilities experience disparities in cancer care? A systematic reviewPLOS ONE

Dear Dr. Tosetti,

Thank you for submitting your manuscript to PLOS ONE. After careful consideration, we feel that it has merit but does not fully meet PLOS ONE’s publication criteria as it currently stands. Therefore, we invite you to submit a revised version of the manuscript that addresses the points raised during the review process.

 Please follow the provided comments to reformat the manuscript to meet the journal requirements.

Please submit your revised manuscript by Oct 08 2023 11:59PM If you will need more time than this to complete your revisions, please reply to this message or contact the journal office at plosone@plos.org. Please include the following items when submitting your revised manuscript:A rebuttal letter that responds to each point raised by the academic editor and reviewer(s). You should upload this letter as a separate file labeled 'Response to Reviewers'.A marked-up copy of your manuscript that highlights changes made to the original version. You should upload this as a separate file labeled 'Revised Manuscript with Track Changes'.An unmarked version of your revised paper without tracked changes. You should upload this as a separate file labeled 'Manuscript'.If applicable, we recommend that you deposit your laboratory protocols in protocols.io to enhance the reproducibility of your results. Protocols.io assigns your protocol its own identifier (DOI) so that it can be cited independently in the future. For instructions see: https://journals.plos.org/plosone/s/submission-guidelines#loc-laboratory-protocols. Additionally, PLOS ONE offers an option for publishing peer-reviewed Lab Protocol articles, which describe protocols hosted on protocols.io. Read more information on sharing protocols at https://plos.org/protocols?utm_medium=editorial-email&utm_source=authorletters&utm_campaign=protocols.

We look forward to receiving your revised manuscript.

Kind regards,

Sina Azadnajafabad, MD, MPH

Academic Editor

PLOS ONE

Journal Requirements:

Additional Editor Comments:

The manuscript needs edits regarding the manuscript formatting and sections of the paper according to the Plos One requirements. Also, the referencing format is not according to the journal style, besides the errors evident in the number of cited manuscript which need a major revision.

Reviewers' comments:

Reviewer's Responses to Questions

**Comments to the Author**

1. If the authors have adequately addressed your comments raised in a previous round of review and you feel that this manuscript is now acceptable for publication, you may indicate that here to bypass the “Comments to the Author” section, enter your conflict of interest statement in the “Confidential to Editor” section, and submit your "Accept" recommendation.

Reviewer #1: All comments have been addressed

2. Is the manuscript technically sound, and do the data support the conclusions?

Reviewer #1: (No Response)

3. Has the statistical analysis been performed appropriately and rigorously? 

Reviewer #1: (No Response)

4. Have the authors made all data underlying the findings in their manuscript fully available?

Reviewer #1: (No Response)

5. Is the manuscript presented in an intelligible fashion and written in standard English?

Reviewer #1: (No Response)

6. Review Comments to the Author

Reviewer #1: Thank you for revising the manuscript according to the comments.

All comments have been adressed in the revision.

7. PLOS authors have the option to publish the peer review history of their article (what does this mean?). If published, this will include your full peer review and any attached files.

Reviewer #1: No

---

## [Author Response · Author response to Decision Letter 1]

10 Nov 2023

Dear Dr. Azadnajafabad,

it is with great pleasure that we resubmit our article for further consideration. We apologize for the long delay in completing this second revision, but unfortunately one of the Authors was absent for several weeks due to sudden health problems. We hope that the responses we provide below satisfactorily address all the issues and concerns that you have noted.

Every section of our paper has been now revised according to Plos One formatting requirements. The referencing format has been changed to the one required by Plos One guidelines, and the numbers of cited manuscripts have been corrected and listed in the proper order.

Thank you again, Dr. Azadnajafabad and Reviewers, for giving us the opportunity to strengthen our manuscript with your valuable comments and queries We have worked to incorporate your feedback and hope that these revisions persuade you to accept our submission.

Dr. Irene Tosetti

---

## [Editor Report · Decision Letter 2]

15 Nov 2023

Do people with disabilities experience disparities in cancer care? A systematic review

PONE-D-23-10494R2

Dear Dr. Tosetti,

We’re pleased to inform you that your manuscript has been judged scientifically suitable for publication and will be formally accepted for publication once it meets all outstanding technical requirements.

Kind regards,

Sina Azadnajafabad, MD, MPH

Academic Editor

PLOS ONE
---

## [Editor Report · Acceptance letter]

17 Nov 2023

PONE-D-23-10494R2 

Do people with disabilities experience disparities in cancer care? A systematic review 

Dear Dr. Tosetti:

I'm pleased to inform you that your manuscript has been deemed suitable for publication in PLOS ONE. Congratulations! Your manuscript is now with our production department. 

Kind regards, 

on behalf of

Dr. Sina Azadnajafabad 

Academic Editor

PLOS ONE